# Tomato Xylem Sap Hydrophobins Vdh4 and Vdh5 Are Important for Late Stages of *Verticillium dahliae* Plant Infection

**DOI:** 10.3390/jof8121252

**Published:** 2022-11-27

**Authors:** Isabel Maurus, Miriam Leonard, Alexandra Nagel, Jessica Starke, James W. Kronstad, Rebekka Harting, Gerhard H. Braus

**Affiliations:** 1Department of Molecular Microbiology and Genetics, Institute of Microbiology and Genetics and Goettingen Center for Molecular Biosciences (GZMB), University of Goettingen, Grisebachstr. 8, D-37077 Goettingen, Germany; 2Michael Smith Laboratories, Department of Microbiology and Immunology, University of British Columbia, 301-2185 East Mall, Vancouver, BC V6T 1Z4, Canada

**Keywords:** hydrophobin, RNA sequencing, *Verticillium dahliae*, virulence

## Abstract

*Verticillium dahliae* causes economic losses to a wide range of crops as a vascular fungal pathogen. This filamentous ascomycete spends long periods of its life cycle in the plant xylem, a unique environment that requires adaptive processes. Specifically, fungal proteins produced in the xylem sap of the plant host may play important roles in colonizing the plant vasculature and in inducing disease symptoms. RNA sequencing revealed over 1500 fungal transcripts that are significantly more abundant in cells grown in tomato xylem sap compared with pectin-rich medium. Of the 85 genes that are strongly induced in the xylem sap, four genes encode the hydrophobins Vdh1, Vdh2, Vdh4 and Vdh5. Vdh4 and Vhd5 are structurally distinct from each other and from the three other hydrophobins (Vdh1-3) annotated in *V. dahliae* JR2. Their functions in the life cycle and virulence of *V. dahliae* were explored using genetics, cell biology and plant infection experiments. Our data revealed that Vdh4 and Vdh5 are dispensable for *V. dahliae* development and stress response, while both contribute to full disease development in tomato plants by acting at later colonization stages. We conclude that Vdh4 and Vdh5 are functionally specialized fungal hydrophobins that support pathogenicity against plants.

## 1. Introduction

Many filamentous fungi produce two to ten hydrophobins, which are small, secreted, moderately hydrophobic proteins. Hydrophobins generally contain eight conserved cysteine residues that form four disulfide bonds; these bonds are important for the solubility of the hydrophobin Sc3 of *Schizophyllum commune*, for example [1]. Hydrophobins are characterized by a high content of hydrophobic amino acids rather than by homology [2,3]. Based on their sequence characteristics, hydrophobicity patterns, and solubility properties, hydrophobins can be divided into two types, class I and class II. Class I hydrophobins, which are found in fungi of the Ascomycota and Basidiomycota phyla, form films that are very insoluble and require strong acids to be dissolved. In contrast, films formed by class II hydrophobins, which are found only in fungi of the Ascomycota, can be dissolved in aqueous organic solvents and detergents [4,5]. Most known hydrophobins accumulate on the surfaces of fungal tissues such as hyphae or spores that are in contact with air or hydrophobic interfaces. Thus, hydrophobins can mediate the attachment of fungi to themselves, symbiotic partners or–in context of pathogenic interactions–to the host surface [1,4,5]. Important roles of hydrophobins in a variety of processes such as fungal development and infection have been suggested for *Aspergillus nidulans*, *Neurospora crassa*, and *Ophiostoma* spp., for example [6,7,8]. In *Magnaporthe oryzae*, hydrophobins are important for conidiation, appressoria formation and pathogenicity on rice [9]. Hydrophobins have been shown to play a general role in the formation of various fungal structures, such as aerial hyphae of *Ophiostoma ulmi* [8] and *S. commune* [10], fruiting bodies of *Cryphonectria parasitica* [11], or microsclerotia of *Verticillium dahliae* [12].

Several fungal species, including *Cladosporium fulvum* [13] and *S. commune* [14], harbor multiple, differentially expressed hydrophobin-encoding genes. In *V. dahliae*, the class II hydrophobin Vdh1 was found to be specifically expressed in developing microsclerotia, hyphal fusions, and conidiophores [15], and to be required for microsclerotia formation [12]. A second *V. dahliae* hydrophobin protein (designated VdHP1, hereafter referred to as Vdh2/VdHP1) exhibits general functions in fungal development and adaptability, such as increased formation of resting structures, increased smoothness of spores, and decreased resistance to abiotic stresses [3]. Vdh1 was initially described as not required for disease development in tomato plants [12]. However, in a second study, the authors reported that constitutive expression of *VDH1* leads to delayed onset of disease symptoms [15]. Negative effects of *V. dahliae* Vdh2/VdHP1 on virulence towards cotton plants have been reported [3].

As a soil-borne pathogen, the ascomycete *V. dahliae* infects its host plants through their roots. Root tips, lateral root hairs and natural root wounds serve as entry ports for the fungal hyphae to colonize the root cortex and central cylinder [16,17,18]. *V. dahliae* forms asexual conidiospores that are spread within the vascular system of the host plant [17,19]. The germinating conidia can eventually block the transport of water and essential minerals through the xylem vessels, leading to the appearance of disease symptoms such as wilting, stunted growth, chlorosis, and early senescence [16,20]. Towards the end of the infection cycle, *V. dahliae* forms melanized resting structures called microsclerotia that are released into the soil from the wilting plant. Microsclerotia allow the fungus to remain viable in the soil for years when environmental conditions are unfavorable or no suitable hosts are available [16,18,21]. After perception of root exudates, the microsclerotia germinate and the hyphae grow towards the plant to initiate another round of infection [16,17]. *V. dahliae* is widespread in both temperate and subtropical regions and can infect a variety of important crops such as cotton, tomato, and sunflower, posing a threat to global crop production [16,18].

We performed RNA sequencing to compare the transcriptomes of *V. dahliae* cultured in xylem sap extracted from tomato plants versus pectin-rich simulated xylem medium (SXM) to identify genes whose expression is specifically upregulated in xylem sap. According to our hypothesis, the proteins derived from the strongly induced genes could play an important role in triggering disease symptoms by facilitating the colonization of the plant vasculature by the fungus. Our RNA sequencing approach revealed that the transcripts of over 1500 genes were significantly more abundant in cells from tomato xylem sap compared with SXM. In addition to the two hydrophobins Vdh1 and Vdh2 mentioned above, two other hydrophobin-encoding genes were strongly expressed (log_2_(fold change) ≥ 4) in the *V. dahliae* wild-type strain JR2 when cultured in xylem sap. These two genes, *VDH4* and *VDH5*, encode hydrophobin proteins that are structurally distinct from the previously described hydrophobins Vdh1 and Vdh2/VdHP1. Our functional investigation revealed that Vdh4 and Vdh5, in contrast to Vdh1 and Vdh2/VdHP1, are specifically important for plant infection without affecting fungal development.

## 2. Materials and Methods

### 2.1. Sample Preparation, RNA Sequencing, and Initial Bioinformatics Analysis

To produce conidiospores, the *V. dahliae* wild-type strain JR2 [22] was grown in liquid SXM (modified from [23] as described by [24]) at 25 °C with shaking at 120 rpm. We harvested spores and separated them from mycelia through sterile Miracloth filters (Calbiochem). Spores were dissolved in sterile water and counted using a Neubauer counting chamber. As a preculture, 50 mL of liquid SXM was inoculated with 5 × 10^7^ spores and incubated at 25 °C for five days with shaking at 120 rpm. Subsequently, mycelia were transferred to either 50 mL of liquid SXM (control) or to 50 mL of extracted tomato xylem sap and incubated at 25 °C for eight h with shaking at 120 rpm. The xylem sap was extracted from six-week-old tomato plants (*Solanum lycopersicum*, ‘Moneymaker’) as previously described [25] and one-tenth volume of sterile distilled water was added. For transfer, the cultures were centrifuged at 2500 rpm for 10 min and the mycelia were washed twice with sterile water. Mycelia were harvested using Miracloth filters and frozen directly in liquid nitrogen. RNA was extracted from the ground material using the TRIzol/chloroform protocol [26]. The Turbo DNA-free Kit (Invitrogen Thermo Fisher Scientific, Waltham, MA, USA) was used for DNase treatment. For transcriptome analysis, three independent replicates were prepared for each culture condition (SXM or xylem sap).

RNA sequencing and initial bioinformatics analysis were performed by Genewiz (Azenta Life Sciences, Chelmsford, MA, USA). The HiSeq platform was used with a 2 × 150 configuration, and polyA selection was performed for eukaryotic mRNA. Low-quality bases and possible adapter sequences were removed using Trimmomatic v.0.36. STAR aligner v.2.5.2b was used to map the trimmed reads to the *V. dahliae* JR2 genome obtained from Ensembl Fungi [27]. Using DESeq2, gene expression was compared between the transcriptomes from the SXM and xylem sap cultures based on unique gene hit counts that were calculated using featureCounts from the Subread package v.1.5.2. Similarities between and within groups were determined by measuring sample distances based on the expression values of each sample (Appendix A). *p*-values and log_2_ fold changes were calculated using Wald tests. Genes with an adjusted *p*-value greater than 0.05 and log_2_(fold change) < 1 were considered not significantly enriched in cells from xylem sap and were excluded from further analyses.

All remaining genes that were significantly higher expressed in cells from xylem sap were analyzed for functional enrichment (Appendix A) with the FungiFun2 web tool [28] using corresponding VdLs.17 identifiers. The web tool can create overlaps between categories so that some genes can be found in multiple categories. Transcripts with a log_2_(fold change) ≥ 4 were considered the most highly induced and were further analyzed separately (Figure 1a, Appendix A) using Ensembl Fungi [27], InterPro [29], DeepLoc [30], BLAST with NCBI [31], and FungiDB [32].

### 2.2. Quantification of Gene Transcription

Gene transcription levels were analyzed by reverse-transcription quantitative PCR using MESA GREEN qPCR MasterMix Plus for SYBR Assay (Eurogentec, Seraing, Belgium) in a CFX Connect Real Time PCR Detection System (Bio-Rad Laboratories, Hercules, CA, USA). Culture conditions were as described above for RNA sequencing. RNAs were extracted from ground mycelia using TRIzol [26]. The QuantiTect Reverse Transcription Kit (Qiagen, Hilden, Germany) was used for cDNA synthesis using 0.8 µg RNA. Transcript levels were quantified in triplicate (*n* = 1) relative to the references histone *H2A* and *EIF2B* using the primers listed in Appendix A. The $2^{-\Delta \Delta C_{T}}$ method was applied [33], and the mean values of three independent experiments ± SE of the mean are shown.

### 2.3. Additional Bioinformatics Methods

Tomato gene sequences were obtained from the Ensembl Plants database, and the *V. dahliae* JR2 gene sequences were obtained from the Ensembl Fungi database [27]. Protein sequences were analyzed using the InterPro website [29]. For statistical analyses, two-tailed Mann–Whitney U-tests [34] were used for plant infection experiments and Welch’s *t*-tests were applied using the iCalcu website (https://www.icalcu.com/stat/two-sample-t-test-calculator.html, accessed in 2021 and 2022) for spore, DNA, or cDNA quantifications (significances: ns, not significant; * *p* < 0.05; ** *p* < 0.01; *** *p* < 0.001; **** *p* < 0.0001).

### 2.4. Plasmid and Strain Construction

The *VDH1* (*VDAG_JR2_Chr2g02500a*), *VDH2* (*VDAG_JR2_Chr4g07990a*), *VDH4* (*VDAG_JR2_Chr3g09420a*), and *VDH5* (*VDAG_JR2_Chr1g24730a*) genes from *V. dahliae* were replaced by resistance marker cassettes via homologous recombination to study the cellular functions of the derived hydrophobin proteins. Single, double and triple deletion strains were generated. DNA fragments were amplified by PCR in Biometra thermal cyclers with Phusion (Thermo Fisher Scientific, Waltham, MA, USA) or Q5 (New England Biolabs, Ipswich, MA, USA) DNA polymerases. The NucleoSpin Gel and PCR Clean-up Kit (Macherey-Nagel, Düren, Germany) was used to purify PCR products. Primers (Appendix A) were designed with 15 base pairs (bp) of homology to the desired neighboring sequences. This allowed assembly of the DNA fragments using the GeneArt Seamless Cloning and Assembly Kit (Thermo Fisher Scientific, Waltham, MA, USA). The resulting plasmids (Appendix A) were used to transform *Escherichia coli* DH5α (Invitrogen Thermo Fisher Scientific, Waltham, MA, USA) using the heat shock method [35,36]. The resulting constructs were confirmed by sequencing (Microsynth Seqlab, Göttingen, Germany). *Agrobacterium tumefaciens* AGL1 [37] was transformed as previously described [38]. *E. coli* and *A. tumefaciens* were cultivated at 37 °C and 25 °C, respectively, in or on lysogeny broth medium [39] supplemented with 100 µg/ mL kanamycin (AppliChem, Darmstadt, Germany) or 100 µg/ mL ampicillin (Carl Roth, Karlsruhe, Germany), as needed for selection. *V. dahliae* mutant strains were generated by *A. tumefaciens*-mediated transformation [40]. Single gene complementation strains were constructed in the double deletion background to verify the observed phenotypes. Potato dextrose medium (PDM; Carl Roth, Karlsruhe, Germany; plus 0.5% agar) supplemented with 72 µg/ mL nourseothricin (Werner BioAgents, Jena, Germany), 50 µg/ mL hygromycin B (InvivoGen, San Diego, CA, USA) and/or 50 µg/ mL geneticin G418 (Merck, Darmstadt, Germany), and 300 µg/ mL cefotaxime (Fujifilm Wako Chemicals, Gunma, Japan) was used for selection as required. All bacterial and fungal strains used in this study are listed in Appendix A.

#### 2.4.1. *VDH2* Deletion Plasmid and Strain Construction

The 407 bp *VDH2* open reading frame (ORF) was replaced by homologous recombination with a 1648 bp geneticin resistance marker under the control of a *trpC* promoter and terminator amplified from pCOM [41] with primers JST253 and JST254. For this purpose, the 1130 bp 5′ flanking region was produced by PCR with primers IM80 and IM81, and primers IM82 and IM83 were used for the 1620 bp 3′ flanking region. Plasmid pME4564 [42], cut with *Eco*RV and *Stu*I, was used as a backbone, and the above-mentioned fragments were inserted (Appendix A). The resulting plasmid pME5484 was used for transformation of *V. dahliae* wild-type to yield *VDH2* deletion strains VGB648 and VGB649.

#### 2.4.2. *VDH4* Deletion Plasmid and Strain Construction

For deletion of *VDH4*, the 510 bp ORF was replaced by a 2194 bp nourseothricin resistance marker under the control of a *gpdA* promoter and a *trpC* terminator amplified from pME4815 [42] with primers ML8 and ML9. The 1100 bp 5′ flanking region was amplified from *V. dahliae* JR2 wild-type genomic DNA using primers IM36 and IM37, while IM38 and IM39 were used to amplify the 900 bp 3′ flanking region. The fragments were inserted into the 6804 bp pME4564 backbone, which was cut with *Eco*RV and *Stu*I (Appendix A). The resulting plasmid was named pME5485 and used to transform the *V. dahliae* wild-type strain, resulting in *VDH4* deletion strains VGB566 and VGB567.

#### 2.4.3. *VDH5* Deletion Plasmid and Strain Construction

*VDH5* was deleted by replacing the 881 bp ORF with a 3942 bp hygromycin B resistance marker under the control of a *gpdA* promoter and a *trpC* terminator amplified with primers ML8 and RO3 from pPK2 [43]. The 1200 bp 5′ flanking region was amplified with primers IM28 and IM29, and the 790 bp 3′ flanking region with primers IM30 and IM31 from wild-type genomic DNA. These fragments were inserted into the *Eco*RV and *Stu*I-cut pME4564 backbone, resulting in pME5486, which was used to transform the *V. dahliae* wild-type strain (Appendix A). The *VDH5* deletion strains were designated VGB568 and VGB569.

#### 2.4.4. Construction of *VDH4/5* Double Deletion Strains

The *VDH4/5* double deletion strains VGB623 and VGB624 were generated by transformation of the *VDH5* deletion strain VGB568 with the *VDH4* deletion plasmid pME5485.

#### 2.4.5. Construction of the *VDH1* Deletion Plasmid and the *VDH1/4/5* Triple Deletion Strains

The 399 bp ORF of *VDH1* was replaced by a 1648 bp geneticin resistance marker under the control of a *trpC* promotor and terminator, which was amplified with primers JST253 and JST254 from pCOM [41]. The 5′ flanking region (900 bp) was amplified with primers IM76 and IM77, and the 3′flanking region (1250 bp) was amplified with primers IM78 and IM79 using wild-type genomic DNA as template. These fragments were inserted into the *Eco*RV and *Stu*I-cut pME4564 backbone, resulting in pME5511, which was used to transform the *V. dahliae VDH4/5* double deletion strain VGB623 (Appendix A). The *VDH1/4/5* deletion strains were named VGB646 and VGB647.

#### 2.4.6. Construction of the *VDH2/5* Double Deletion and *VDH2/4/5* Triple Deletion Strains

Transformation of the *VDH5* deletion strain VGB658 with the *VDH2* deletion plasmid pME5484 resulted in a *VDH2/5* double deletion strain, which was designated VGB651. The *VDH2/4/5* triple deletion strain VGB657 was generated by transformation of the *VDH4/5* double deletion strain VGB623 with pME5484.

#### 2.4.7. Construction of *VDH4* and *VDH5* Single Gene Complementation Strains in the *VDH4/5* Double Deletion Background

*VDH4* and *VDH5* single gene complementation constructs were introduced separately into the *VDH4/5* double deletion strain to verify the observed phenotypes. For the *VDH4* complementation construct, the 5′ flanking region, the backbone of pME4564, and the 3′ flanking region were amplified from pME5485 using primers ML184 and ML185. *VDH4* was amplified from wild-type genomic DNA using primers IM54 and ML186. The 1648 bp geneticin resistance marker cassette was amplified from pCOM [41] using primers JST253 and JST254. All fragments were ligated (Appendix A) and the resulting plasmid was conserved as pME5487. This plasmid was used to transform the *VDH4/5* double deletion strain, resulting in the *VDH4*-C/Δ*VDH5* strains VGB690 and VGB691.

The *VDH5* complementation construct was generated by amplifying the 5′ and 3′ flanking regions and the pME4564 backbone from pME5486 with primers ML187 and ML188. *VDH5* was amplified from wild-type genomic DNA using primers IM52 and ML189. These fragments were ligated together with the geneticin resistance marker cassette (Appendix A), and the resulting plasmid pME5488 was used to transform the *VDH4/5* double deletion strain. The resulting *VDH5*-C/Δ*VDH4* strain was designated VGB692.

### 2.5. Isolation of Genomic DNA and Southern Hybridization

For isolation of genomic DNA, *V. dahliae* strains were grown with shaking at 120 rpm and 25 °C in liquid PDM. Mycelia were harvested using Miracloth and ground to powder in liquid nitrogen. Genomic DNA was isolated with phenol as described previously [42]. Southern hybridizations were performed as described [40,42] using Amersham AlkPhos Direct Labelling and CDP-Star detection reagents (GE Healthcare, Chicago, IL, USA).

### 2.6. Phenotypical Analyses

Colony growth and microsclerotia formation on media containing 2% agar were studied after point-inoculation of 5 × 10^4^ spores and incubation at 25 °C for ten days. PDM, SXM, and Czapek-Dox medium (CDM, modified from [44] as described by [45]) were used. Morphology under different stress conditions was tested by replacing sucrose in CDM with 3% cellulose or supplementing CDM with 0.8 M sorbitol, 0.5 M sodium chloride, 0.004% SDS, or 0.000075% hydrogen peroxide. *Ex-planta* phenotypes were examined by binocular microscopy (SZX12-ILLB2-200, illuminated with KL1500-LCD light source and equipped with SC30 camera, Olympus, Tokyo, Japan) with cellSens Dimension software (Olympus, Tokyo, Japan).

### 2.7. Quantification of Conidiospore Formation

The amount of conidia formed by the strains was determined according to the protocol described previously [26,46]. Briefly, each strain was inoculated in three to four technical replicates (*n* = 1). Cultures were incubated for five days with shaking. Spores were harvested using Miracloth and counted using the Coulter Z2 Particle Count and Size Analyzer (Beckman Coulter, Brea, CA, USA) with Beckman Coulter Isoton II Diluent.

### 2.8. Infection of Tomato Plants

*S. lycopersicum* (‘Moneymaker’, Kiepenkerl Bruno Nebelung, Everswinkel, Germany) seeds were surface-sterilized with 70% (*v/v*) ethanol and 0.05% Tween 20. Subsequently, ten-day-old seedlings were wounded and inoculated with spores or water (mock) by root-dipping as previously described [42,45]. After 21 days of incubation under long-day conditions in a BrightBoy GroBank (CLF Plant Climatics, Wertingen, Germany) discoloration of tomato hypocotyls was observed by binocular microscopy.

The amount of fungal DNA in hypocotyls was determined by quantitative PCR. Genomic DNA was isolated using the NucleoSpin Plant II Kit (Macherey-Nagel, Düren, Germany) with pooled hypocotyls inoculated with the individual strains or mock-inoculated. SsoAdvanced Universal SYBR Green Supermix (Bio-Rad Laboratories, Hercules, CA, USA) and primers OLG70 and OLG71 that amplify a specific *Verticillium* fragment of the *5.8S rRNA* [17] were used to test 10 ng of DNA each in technical triplicates. The $2^{-\Delta \Delta C_{T}}$ method [33] was applied. The tomato elongation factor-encoding gene *EF1α* and wild-type *V. dahliae* DNA were used for normalization. Shown are the means of four biological replicates ± SE of the mean.

Total disease scores of tomato plants are presented in stacking diagrams. Disease symptoms were determined by measuring plant height, weight and the length of the longest leaf and converted to a disease score ranking relative to the mean values of mock-inoculated plants (set to 100%). Plants with values above 80% were classified as ‘healthy’, values of 60–80% resulted in classification as ‘mild symptoms’, 40–60% as ‘strong symptoms’ and below 40% as ‘very strong symptoms’.

## 3. Results

### 3.1. Expression of 85 V. dahliae Genes, Including Four Hydrophobin Genes, Is Strongly Induced in Tomato Xylem Sap

Pectin-rich SXM was originally developed to mimic plant xylem sap [23]. However, secretome analyses revealed that *V. longisporum* produces a specific secretion pattern when grown in SXM, which partially differs from responses observed after cultivation in xylem sap extracted from the host plant rapeseed [42]. These results suggest that SXM is not able to fully mimic the growth condition in pure xylem sap. Therefore, we compared the transcriptome of *V. dahliae* wild-type strain JR2 grown in SXM versus cultivation in xylem sap obtained from uninfected tomato plants to identify genes whose expression is specifically induced in xylem sap. Overall, the expression of 1533 genes was significantly higher in the tomato xylem sap condition compared with SXM. The FungiFun2 web tool [28] was used for functional enrichment analysis. With this approach, 448 genes were found in 15 significantly enriched categories (Appendix A). In summary, genes were attributed functions in the main areas of metabolism, cellular transport, virulence, and interactions with the environment. The web tool generates overlap between categories thus resulting in some genes being found in multiple categories. About 200 candidates were assigned a function in secondary metabolism and more than 200 had functions in amino acid or carbohydrate metabolism. Among the latter functions, we identified *VDAG_JR2_Chr6g04040a* encoding a xylanase previously characterized in *V. dahliae* Vd991. This xylanase (VdXyn4) has been shown to be the only one, among six xylanases encoded in the Vd991 genome, that degrades the plant cell wall and contributes to virulence by acting at late stages of infection after pathogen entry into the xylem vessels [47]. Our analysis also revealed 50 candidates found to be involved in ion transport, 90 that were assigned to amino acid or carbohydrate transport, and 21 related to allantoin transport. Allantoin is an important N-containing compound that is transported in the xylem stream and that can be used as a carbon and N-source by fungi and plants [48,49]. Virulence associations were found for 33 candidates and an additional 40 were connected to interactions with the environment.

The inferred proteins of the genes most strongly induced in the xylem sap condition with a log_2_(fold change) ≥ 4 were examined in more detail. For this purpose, functions were manually assigned based on the identification of conserved domains and/or characterized putative homologs by BLAST searches (Appendix A). The expression of 85 *V. dahliae* genes was strongly upregulated upon culture in tomato xylem sap. The identified candidates are believed to have functions in hydrophobicity, metabolism, proteolysis, redox processes, transcription, transport, and in other processes (Figure 1a). Approximately half of the putative metabolic candidates were predicted to be involved in carbohydrate metabolism. Additionally, many candidates related to sugar transport were identified within the functional category ‘transport.’ Among the candidates assigned to other processes, one gene (*VDAG_JR2_Chr7g01630a*) encodes a protein with a predicted domain of the pectin lyase fold/virulence factor superfamily. A similar protein of *V. dahliae* Vd991, which belongs to the same superfamily, was shown to induce severe cell death in several plants and to contribute to virulence [50]. Another gene (*VDAG_JR2_Chr2g04040a*) encodes a protein putatively involved in adhesion. Adhesive proteins are important at several stages of host-pathogen interactions [51]. No functional category could be assigned for 23 transcripts because of the lack of characterized homologs and conserved domains in the derived proteins. Strikingly, more than one third (30 of 85) of the identified candidates are predicted to contain an N-terminal signal peptide that targets the derived proteins for secretion. Among them, one derived protein (VDAG_JR2_Chr4g11330a) met the requirements for a potential effector (≤ 200 amino acids (aa), predicted N-terminal signal peptide, ≥ 2% cysteines [52]). In addition, DeepLoc [30] predicted localization in the extracellular space for seven further proteins, although no signal peptide was predicted for these candidates (Appendix A). Of the four predicted hydrophobin proteins, three are thought to contain a signal peptide, while the fourth hydrophobin was predicted to have an extracellular localization.

These results highlight that the fungus *V. dahliae* can adapt its gene expression to the environment, producing specific patterns in xylem sap that differ from pectin-rich medium.

### 3.2. Vdh4 and Vdh5 Are Distinctive Members among the Five Hydrophobins Annotated in V. dahliae JR2

Our RNA sequencing approach revealed that expression of the four hydrophobin-encoding genes *VDH1* (*VDAG_JR2_Chr2g02500a*), *VDH2* (*VDAG_JR2_Chr4g07990a*), *VDH4* (*VDAG_JR2_Chr3g09420a*), and *VDH5* (*VDAG_JR2_Chr1g24730a*) was strongly upregulated when *V. dahliae* was cultured in extracted tomato xylem sap compared with SXM (Figure 1a, Appendix A). In the genome of *V. dahliae* JR2, five genes are annotated that encode proteins carrying a predicted cerato-ulmin hydrophobin family domain (IPR010636) at the C-terminal end (Figure 2). Transcription patterns of the five hydrophobin-encoding genes were verified by reverse-transcription quantitative PCR. The transcript levels of *VDH1*, *VDH2*, *VDH4*, and *VDH5* were significantly higher in cells from xylem sap compared with SXM, whereas no transcript of *VDH3* was amplified under either condition (Figure 1b).

Based on cysteine spacing patterns (according to [53]) all five hydrophobin proteins were classified as class II hydrophobins (Appendix A). Further analysis of the deduced proteins using InterPro [29] and Ensembl Fungi [27] revealed that Vdh1, Vdh2, and Vdh3 (VDAG_JR2_Chr1g24880a) are very similar proteins with 95–99 aa and a predicted molecular weight of 9.3–10.4 kDa. In addition to the cerato-ulmin hydrophobin family domain, these three proteins each carry a predicted N-terminal signal peptide. While Vdh1 [12] and Vdh2/VdHP1 [3] have been functionally characterized, this is not the case for Vdh3. In *V. dahliae* Dvd-T5, Vdh1 was required for microsclerotia production [12]. Recently, Vdh2/VdHP1 has been associated with various functions in the development, adaptability, and virulence of *V. dahliae* Vd080 isolated from cotton [3]. In our experiments, the transcript of the Vdh3-encoding gene was not found during cultivation in SXM or xylem sap (Figure 1b). In contrast, transcripts of two other hydrophobin-encoding genes, *VDH4* and *VDH5*, were found in high abundance during cultivation of *V. dahliae* JR2 in plant xylem sap (Figure 1b, Appendix A). The derived proteins Vdh4 and Vdh5 differ from the group formed by Vdh1, Vdh2, and Vdh3 (Figure 2). Vdh4 is the only predicted hydrophobin of *V. dahliae* JR2 that is not thought to carry a signal peptide at its N-terminal end but was nevertheless predicted by DeepLoc [30] to be an extracellular protein (probability: 0.95). Vdh4 consists of 134 aa with a molecular weight of 13.8 kDa, whereas Vdh5 is an even larger protein with 250 aa and a predicted molecular weight of 23.5 kDa. Originally, Vdh5 was annotated as a 258 aa protein, which we corrected to 250 aa by cDNA sequencing because of the presence of an additional unpredicted intron. The cDNA sequence of *VHD5* and the amino acid sequence of the deduced protein Vdh5 are depicted in Appendix A, respectively. The sequence of Vdh5 is extended at the N-terminus by glycine and asparagine repeats. In addition to the predicted N-terminal signal peptide and C-terminal domain of the cerato-ulmin hydrophobin family, Vdh5 is the only *V. dahliae* JR2 protein predicted to have a dense granule Gra6 protein domain (IPR008119). The Gra6 protein, encoded by a single copy gene in *Toxoplasma gondii*, has been linked to the parasitophorous vacuole of this obligate protozoan parasite that infects many vertebrate hosts [54,55].

In light of the sequence and expression information, the functions of the two unique hydrophobins Vdh4 and Vdh5 in the life cycle and virulence of *V. dahliae* JR2 were analyzed further, and compared with Vdh2 as a representative of the structurally similar Vdh1, Vdh2, and Vdh3 proteins (Figure 2).

### 3.3. Hydrophobins Vdh2, Vdh4, and Vdh5 Are Dispensable for V. dahliae ex Planta Growth, Microsclerotia Formation, Conidiation, and Stress Response

Single gene deletion strains of *VDH2*, *VDH4*, and *VDH5* in the *V. dahliae* JR2 strain background were generated by replacing the respective ORF with a geneticin, nourseothricin, or hygromycin resistance marker, respectively (Appendix A). Furthermore, double deletion strains of *VDH2/5* and *VDH4/5* as well as triple deletion strains of *VDH1/4/5* (Appendix A) and *VDH2/4/5* were constructed. In-locus complemented strains were generated by separately reintroducing *VDH4* and *VDH5* into the *VDH4/5* double deletion strain (Appendix A). Correct integration of the deletion or complementation constructs was confirmed by Southern hybridizations (Appendix A).

We investigated a possible function of hydrophobins in vegetative growth and microsclerotia formation outside the plant host by point-inoculating spores on PDM, SXM and CDM. Melanized microsclerotia are the resilient resting structures that allow *V. dahliae* to survive in nature for long periods of time [16,18,21]. In *V. dahliae* Dvd-T5, expression of *VDH1* is required for the formation of microsclerotia [12]. In our experiments, neither the *VDH4* or *VDH5* single deletion strains nor the *VDH4/5* double deletion strain showed any phenotypic differences from the wild-type (Figure 3a). The *VDH4* and *VDH5* complementation strains in the *VDH4/5* double deletion strain background also grew to wild-type-like size and shape, and no difference in melanization was observed. In addition, growth and microsclerotia formation in the absence of *VDH2* were examined. The *VDH2* single deletion, *VDH2/5* double deletion, and the *VDH2/4/5* triple deletion strains were phenotypically indistinguishable from wild-type. (Appendix A). Vdh1 of the *V. dahliae* JR2 isolate also appears to be dispensable for growth and microsclerotia formation, because *VDH1/4/5* triple deletion strains resembled the wild-type strain on PDM, SXM and CDM plates (Appendix A).

A possible involvement of hydrophobins Vdh2, Vdh4, and Vdh5 in stress responses was also tested. For this purpose, cellulose as an alternative carbon source, sorbitol or sodium chloride for osmotic stress, SDS as a cell wall-perturbing agent, or the oxidative stressor hydrogen peroxide were added to CDM plates. Growth and microsclerotia formation were unaffected by the gene deletions in all single, double, and triple deletion strains under any of the stress-inducing conditions (Appendix A).

*V. dahliae* produces asexual spores (conidia) on conidiophores and, as a vascular pathogen, uses conidia to spread throughout the host plant via the xylem sap stream [16,18,19]. We analyzed conidia production in the absence of *VDH4* and *VDH5* in liquid SXM and compared it with wild-type. Conidiation was wild-type-like in the single and double deletion strains (Figure 3b). In addition, a possible involvement of Vdh2 in asexual spore production was tested. The *VDH2* single deletion, *VDH2/5* double deletion, and the *VDH2/4/5* triple deletion strains formed conidia at wild-type levels (Appendix A).

Taken together, these results suggest that hydrophobins Vdh1, Vdh2, Vdh4, and Vdh5 are dispensable for the growth and development of *V. dahliae* JR2 outside the plant.

### 3.4. Hydrophobins Vhd4 and Vdh5 Are Not Required for Initial Colonization of Tomato Plants

Possible functions of Vdh4 and Vdh5 at different stages of plant infection were investigated. For this purpose, the roots of ten-day-old tomato seedlings were treated with *V. dahliae* spores or water as control (mock). As a soil-borne pathogen, *V. dahliae* invades the vascular tissue through the roots of the host plant [16]. One of the first signs of fungal infection is the discoloration of the plant hypocotyl visible in cross-sections. We observed brownish streaks in hypocotyls of plants infected by *VDH4* and *VHD5* single and double deletion strains, and likewise in JR2 wild-type-infected plants (Figure 4a). Infection by the *VDH4* and *VDH5* single gene complementation strains in the *VDH4/5* double deletion background similarly resulted in hypocotyl discolorations, whereas no discoloration was visible in hypocotyls of mock-inoculated plants.

Fungal DNA in hypocotyls was quantified relative to plant DNA to determine the presence of *V. dahliae* and possible changes in initial colonization stages. By quantitative PCR, less fungal DNA was measured only in mock-inoculated plants. No significant differences were found in the amount of fungal DNA between wild-type-infected plants and plants inoculated with the single and double deletion and complementation strains of *VDH4* and *VDH5* (Figure 4b). In contrast, when tomato plants were infected by *V. dahliae VDH2* deletion strains, less fungal DNA was detected in hypocotyls compared with wild-type-infected plants, although hypocotyl discoloration was visible in all plants infected by the single, double, and triple deletion strains (Figure 5).

Taken together, these data suggest that Vdh4 and Vhd5, unlike Vdh2, are not necessary to reach the plant hypocotyl from the roots and thus are not required for the initial steps of tomato plant colonization by *V. dahliae*.

### 3.5. The Presence of Either VDH4 or VDH5 Is Required for Wild-Type-like Disease Development in Tomato Plants

Expression of hydrophobin-encoding genes is strongly induced when *V. dahliae* JR2 is cultured in xylem sap extracted from the host plant tomato compared with SXM (Figure 1, Appendix A). We therefore investigated whether Vdh1, Vdh2, Vdh4, and Vdh5 are involved in advanced stages of systemic plant infection and symptom expression in tomato plants. Disease symptoms were assessed at 21 days after seedlings were inoculated with spores from single, double, and triple deletion strains as well as complementation strains. A general disease index was calculated from the measured plant height, longest leaf length, and plant weight [42,45]. Plants were classified as healthy or with weak, strong, or very strong symptoms based on this score when compared with water-treatment (mock). Single deletion of either *VDH4* or *VDH5* did not in all cases result in less severe disease symptoms in tomato plants. For example, single deletion of *VDH4* or *VDH5* in the JR2 wild-type background led to approximately 15–20% more plants with no or only weak symptoms. The same was true for the *VDH4* single gene complementation strain, and reintroduction of *VDH5* into the *VDH4/5* double deletion background resulted in disease symptoms comparable with those of wild-type infection. Simultaneous deletion of *VDH4* and *VDH5* resulted in lower infection efficiency. Compared with wild-type-infected plants, about 25% more plants displayed no or only weak symptoms (Figure 6).

Vdh1 and Vdh2 did not contribute to disease development in tomato plants. Infection with *VDH2* single deletion and *VDH2/5* double deletion strains led to wild-type-like disease outcomes (Figure 7), although fungal DNA levels appeared to be reduced in hypocotyls (Figure 5b). As observed with the *VDH4/5* double deletion strain, root inoculation with spores of the *VDH2/4/5* triple deletion strain resulted in 25% more plants with only weak symptoms compared with wild-type-infected plants. The additional deletion of *VDH1* in the *VDH4/5* deletion strain caused inoculated tomato plants to exhibit symptoms comparable to those of plants inoculated with the double deletion strain (Appendix A).

Overall, these data suggest that the hydrophobins Vdh4 and Vdh5, but not Vdh1 and Vdh2, contribute to full disease development in tomato plants by acting at later colonization stages.

## 4. Discussion

The two hydrophobins Vdh4 and Vdh5 of *V. dahliae* JR2 each have a particular structural composition, which is important in the xylem sap of tomato plants for enhancing disease symptoms. We found that the expression of the corresponding genes is greatly increased when *V. dahliae* is grown in xylem sap from tomato plants compared with pectin-rich medium. The pectin-rich medium was designed to mimic growth conditions in plant vasculature; however, previous studies have shown that *Verticillium* spp. can differentiate between natural and artificial media [42].

Xylem sap of rapeseed plants has been shown to inhibit the growth of *V. longisporum*, while cultivation in filtered xylem sap, which excludes macromolecules larger than 3 kDa, enhanced fungal growth. In filtered xylem sap, greater numbers of the spores required for dispersal in the xylem vessels are produced [56]. These results indicate that plant signal molecules larger than 3 kDa are present in unfiltered xylem sap and are sensed by the fungus, triggering adaptation of its growth and development. The expression of fungal proteins does not change significantly in the presence of either uninfected or infected xylem sap [56], suggesting that plant signal molecules are present in the xylem sap regardless of the infection status of the plant. In our RNA sequencing approach, uninfected xylem sap from the *V. dahliae* host plant tomato was used and not filtered to better resemble the natural situation in the plant. We found that the transcript levels of over 1500 fungal genes are significantly higher during growth in xylem sap compared with SXM. It is possible to identify previously characterized virulence determinants using this approach. For example, this study found a gene encoding a xylanase (Appendix A) that has already been shown to contribute to late stages of infection when *V. dahliae* has invaded xylem vessels [47].

Xylem sap provides a physically unique, relatively nutrient-poor habitat for fungal growth within the plant [57,58,59]. Genes encoding proteins involved in carbohydrate metabolism and transport of sugars and amino acids are found in the significantly enriched categories (Appendix A), suggesting that *V. dahliae* senses and responds to the composition of its environment by adjusting gene expression accordingly to obtain sufficient nutrients for growth and reproduction. For *V. longisporum*, it was shown that the fungus adapts to the limited sugar and unbalanced amino acid supply in xylem sap by activating the cross-pathway control transcription factor Cpc1. In the same study, deletion of *CPC1* in *V. dahliae* resulted in high sensitivity to amino acid deficiency and lower symptom expression in tomato plants [60].

Previously, RNA sequencing or microarray analyses have been used to analyze *V. dahliae* genes expressed in (germinating) microsclerotia [61,62,63,64] and in response to nutrient deficiency [65], or to study gene regulation by transcription factors [25,66,67]. Transcriptomic studies of *Verticillium*-plant interactions also often focus on sequencing of plant RNAs [68,69,70,71,72]. Analyses on the fungal transcriptome interacting with the plant host are limited and obtaining proteomic data from *in-planta* experiments is challenging because fungal protein concentrations are low compared with plant proteins. Transcriptional changes in early stages of infection were investigated for both *V. dahliae* and the model plant *Arabidopsis thaliana* after 24 h of co-cultivation [73]. Comparable to our data, the study showed the upregulation of many genes for sugar metabolism and sugar transport processes. This suggests that *V. dahliae* reprograms its primary metabolism quite early after perception of the plant. Another transcriptome study of *V. dahliae* focused on the differential responses towards susceptible and tolerant olive cultivars, and found that fungal genes involved in niche-adaptation and virulence were preferentially induced in the susceptible cultivar [74]. Our approach focused on the fungal transcriptome during contact with the plant environment. The expression of 85 of the genes identified in this study was strongly induced in xylem sap (Figure 1, Appendix A). Among them, we found four genes coding for hydrophobins. In a transcriptome analysis comparing a *V. dahliae* strain with low virulence on cotton with a closely related highly virulent one, five hydrophobin-encoding genes were found to be upregulated in the high-virulence strain [75]. This result suggests that hydrophobins may be important for the fungal interaction with plants. Diverse functions in developmental processes have been described for this group of proteins. They are found on the outer surfaces of fungal structures, where they mediate interactions between the fungus and the environment [76,77], suggesting a possible connection to virulence.

Loss-of-function mutation of *VDH1* in the *V. dahliae* Dvd-T5 isolate was found to decrease the number of microsclerotia and the desiccation tolerance of spores. However, Vdh1 is not required to produce normal conidiophores and spores, or for disease development in tomato plants [12]. Vdh1 likely mediates the development of microsclerotia, as the gene is specifically expressed in developing microsclerotia, hyphal fusions, and conidiophores. Constitutive expression of *VDH1* leads to delayed onset of disease symptoms without affecting in vitro microsclerotia development [15]. A second *V. dahliae* hydrophobin protein Vdh2/VdHP1 has been characterized in the Vd080 isolate. As with Vdh1, Vdh2/VdHP1 is necessary for microsclerotia formation. Furthermore, Vdh2/VdHP1 increases spore smoothness while decreasing resistance to abiotic stresses [3]. Overall, hydrophobins in *V. dahliae* have been previously associated mainly with functions in developmental processes, especially the formation of resting structures, and with different responses to various stress factors. In our experiments, none of the tested deletion strains lacking the hydrophobin-encoding genes *VDH1*, *VDH2*, *VDH4*, or *VDH5* showed any alteration in growth, microsclerotia and spore formation, or stress response (Figure 3, Appendix A). Thus, our results suggest that the developmental functions of hydrophobins depend on the isolate used. Strain and host specificity have also been suggested for Nep1-like proteins [42,78,79], the cerato-platanin domain-containing protein Cp1 [42,80], and the polyketide synthase Pks1 [81,82].

The defects in fungal development caused by the deletion of hydrophobin-encoding genes may also affect the infection process in some cases. For example, there has been controversy over a class II hydrophobin from *O. ulmi* possibly contributing to the spread of Dutch Elm disease by promoting adhesion of the fungus to the insect vector [83]. Two hydrophobin gene deletion strains of *M. oryzae* have defects in appressorium and spore development, which are linked to pathogenicity [9,84]. Vdh2/VdHP1 of *V. dahliae* Vd080 has also been associated with functions in Verticillium wilt of cotton by inducing cell death and plant immune responses, and negatively affecting virulence on cotton plants [3]. Our experiments revealed no involvement of Vdh2 in the virulence of *V. dahliae* JR2. When tomato plants were infected by *V. dahliae VDH2* deletion strains, less fungal DNA was detected in tomato hypocotyls compared with wild-type-infected plants (Figure 5). However, the reduced fungal biomass in the hypocotyl did not alter the observed plant disease symptoms (Figure 7). In contrast, deletion of *VDH4* and *VDH5* results in the detection of fungal DNA levels in infected hypocotyls comparable with those of the wild-type (Figure 4), suggesting that these two hydrophobins are dispensable for initial colonization of tomato plants starting from the roots.

This study focused mainly on the two hydrophobin proteins Vdh4 and Vdh5. Vdh4 is the only *V. dahliae* hydrophobin that presumably does not contain a signal peptide (Figure 2). Nevertheless, extracellular localization is predicted. Proteins lacking signal peptides can be exported via Golgi-independent, unconventional secretion systems [85]. The ability of *V. dahliae* to export proteins via systems other than the conventional secretory pathway is supported by exoproteomic studies [86,87]. Vdh5 is a larger protein and contains an N-terminal extension harboring a dense granule Gra6 domain (Figure 2). Most class II hydrophobins are small and contain few structural features other than the core structure, which consists of four beta-sheets and a single helix [88]. Nevertheless, some hydrophobin proteins have been identified that have an elongated N-terminus. This region is characterized by a conserved repeat of glycine and asparagine. Similar repeats are found in a yeast protein that can form stabilized filaments [89]. Therefore, it is possible that the extended N-terminus also forms defined structures that contribute to the structural rigidity of hydrophobins [5].

Interestingly, we identified Vdh4 and Vdh5 as novel and specific virulence factors during the late stages of tomato plant infection by *V. dahliae* JR2. The two hydrophobins are each dispensable for initial colonization of tomato hypocotyls (Figure 4). Yet, the expression levels of *VDH4* and *VDH5* are highly elevated in tomato xylem sap (Figure 1b, Appendix A). The hydrophobin-encoding gene *MPG1* is strongly expressed late in the disease cycle of *M. oryzae*, coinciding with the appearance of symptoms on rice plants. Secretion of large amounts of hydrophobin proteins could possibly inhibit xylem functions [9]. Deletion of *MHP1*, another *M. oryzae* hydrophobin-encoding gene that is highly expressed during plant colonization, results in reduced infectious growth [84]. It can be speculated that hydrophobins might be necessary for spore attachment in the plant for germination and penetration into adjacent tissues, leading to completion of the disease cycle and full pathogenicity [84]. Deletion of *VDH4* or *VDH5* alone already results in less severe disease symptoms on tomato plants, but not in all cases. However, simultaneous deletion of *VDH4* and *VDH5* consistently results in milder infection symptoms (Figure 6). This suggests that Vdh4 and Vdh5 have at least partially overlapping functions that contribute to the virulence of *V. dahliae*, and that the presence of either Vdh4 or Vdh5 is required for strong symptom expression in tomato plants. While Vdh1 of *V. dahliae* Dvd-T5 is important for microsclerotia formation [12] and Vdh2/VdHP1 of Vd080 is involved in fungal development, adaptability and virulence on cotton plants [3], the hydrophobins Vdh1, Vdh2, Vdh4, and Vdh5 of JR2 are dispensable for development. Therefore, overall, our results suggest that hydrophobins might play different roles in development and virulence depending on the isolate and host plant.

In conclusion, RNA sequencing of *V. dahliae* grown in xylem sap can be used to identify factors important for vascular colonization and virulence of the fungal plant pathogen. Using our approach, four hydrophobin-encoding genes were discovered whose expression is strongly induced in extracted xylem sap. Two of the deduced proteins, Vdh4 and Vdh5, are special hydrophobins in terms of both structure and function. Unlike other described hydrophobins, Vdh4 and Vdh5 of *V. dahliae* JR2 are not involved in the development of fungal structures, such as microsclerotia or spores. Both are specifically important in advanced stages of plant infection by *V. dahliae*, where they serve at least partially overlapping functions. In the future, cytological studies comparing mutant strains with wild-type during plant colonization may provide further insights into the functions of hydrophobins in host interactions. Understanding the molecular basis of the newly discovered function of Vdh4 and Vdh5 in plant disease may help combat the increasing threat of Verticillium wilt to our crops.

## Figures and Tables

**Figure 1 jof-08-01252-f001:**
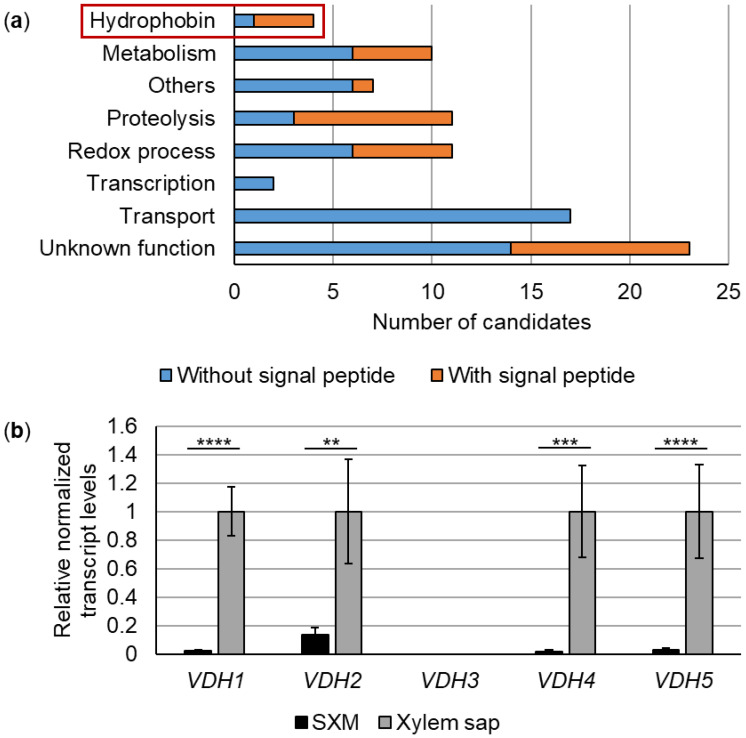
Expression of 85 *Verticillium dahliae* genes, including four genes encoding hydrophobins, is strongly induced in tomato xylem sap. Wild-type spores of strain JR2 were used to inoculate liquid simulated xylem medium (SXM) and incubated for five days with shaking. Mycelia were then transferred to either extracted tomato xylem sap or SXM and incubated for an additional eight h. (**a**) RNA sequencing was performed on three biological replicates per condition. After applying a log_2_(fold change) cut-off of ≥ 4, transcript levels of 85 genes were significantly higher in cells from extracted tomato xylem sap compared with SXM. InterPro and BLAST were used to assign putative functions based on conserved domains. The products of the differentially expressed genes include hydrophobin proteins (red box), transcription factors, transporters, proteins involved in metabolism, proteolysis and redox processes, and proteins with other or unknown functions. Proteins lacking a signal peptide are shown in blue, whereas proteins containing a predicted signal peptide are shown in orange. Details are provided in Appendix A. (**b**) The transcription of hydrophobin-encoding genes was analyzed by reverse transcription-quantitative PCR. Normalization to the transcript levels of the reference genes *H2A* and *EIF2B* was performed. Transcript levels for the data from the xylem sap condition were set to one. Mean values from three independent experiments are shown. Significance was calculated using Welch’s *t*-tests (**, *p* < 0.01; ***, *p* < 0.001; ****, *p* < 0.0001). The transcript levels of *VDH1*, *VDH2*, *VDH4,* and *VDH5* were significantly higher in the xylem sap versus the SXM condition. The transcript of *VDH3* was not amplified in either the SXM or xylem sap conditions. The error bars represent the standard errors of the mean.

**Figure 2 jof-08-01252-f002:**
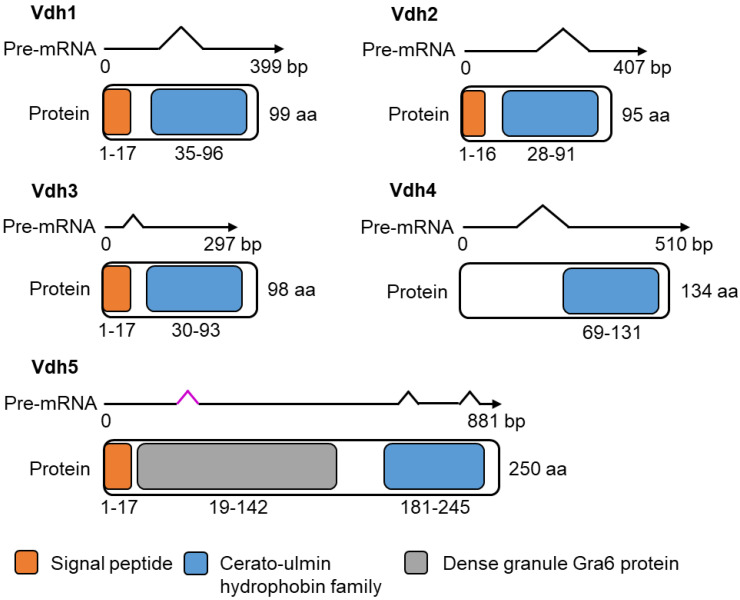
Transcript and protein structure of *V. dahliae* hydrophobins. All five hydrophobins annotated in JR2 (Vdh1: VDAG_JR2_Chr2g02500a, Vdh2: VDAG_JR2_Chr4g07990a, Vdh3: VDAG_JR2_Chr1g24880a, Vdh4: VDAG_JR2_Chr3g09420a, and Vdh5: VDAG_JR2_Chr1g24730a) contain a C-terminal domain of the cerato-ulmin hydrophobin family (IPR010636; in blue) predicted by InterPro. Vdh1, Vdh2, Vdh3, and Vdh5, but not Vdh4, additionally contain a predicted N-terminal secretory signal peptide (in orange). Vdh1, Vdh2, and Vdh3 consist of 95 to 99 amino acids (aa) encoded by open reading frames (ORF) of 297 to 407 base pairs (bp). Vdh4 is slightly larger with 134 aa encoded by a 510 bp ORF. Vdh5 represents the largest protein with 250 aa encoded by an 881 bp ORF and has an additional predicted dense granule Gra6 protein domain (IPR008119; in gray). The original Vdh5 annotation of 258 aa was corrected to 250 aa by cDNA sequencing because of an additional unpredicted intron (highlighted in purple). The cDNA sequence of *VDH5* and the amino acid sequence of the deduced protein are shown in Appendix A, respectively.

**Figure 3 jof-08-01252-f003:**
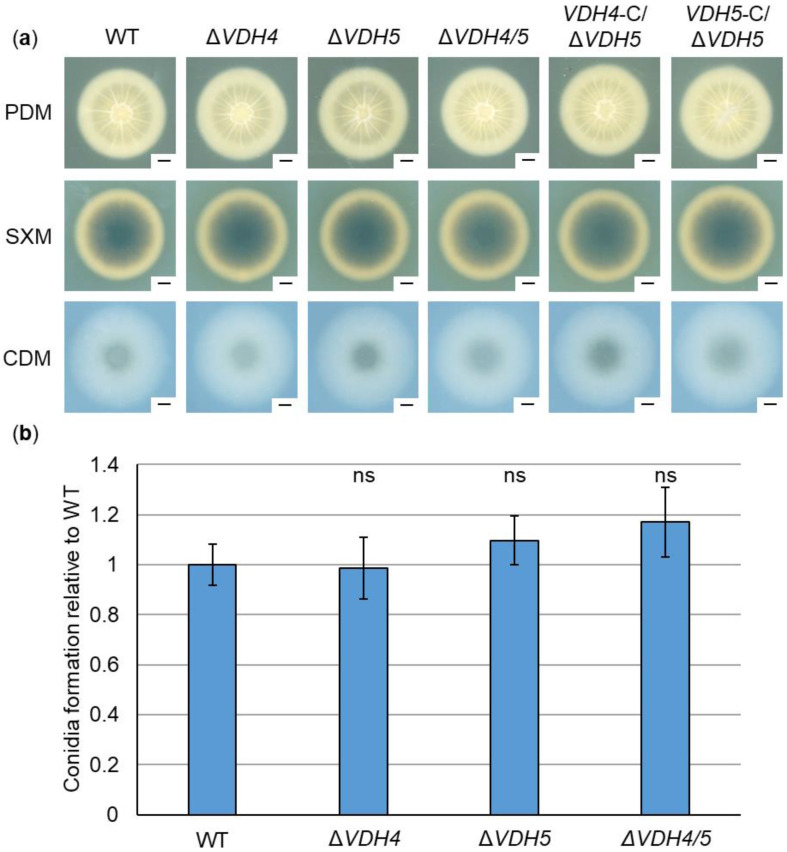
*Ex-planta* growth, microsclerotia formation and conidiation of *V. dahliae* do not depend on the presence of hydrophobin-encoding genes *VDH4* and *VDH5*. JR2 wild-type (WT) was compared with *VDH4* (Δ*VDH4*) and *VDH5* (Δ*VDH5*) single deletion strains, *VDH4/5* double deletion strain (Δ*VDH4/5*), and *VDH4* (*VDH4*-C/Δ*VDH5*) or *VDH5* (*VDH5*-C/Δ*VDH4*) single gene complementation strains in the double deletion strain background. (**a**) Potato dextrose medium (PDM), simulated xylem medium (SXM) and Czapek-Dox medium (CDM) plates were point-inoculated with 50,000 spores of the respective strains and incubated at 25 °C for ten days. Scans from below depict similar morphology of wild-type, Δ*VDH4,* Δ*VDH5*, Δ*VDH4/5*, *VDH4*-C, and *VDH5*-C on all media tested (scale = 0.5 cm). (**b**) Conidia formation was quantified in five-day-old cultures in liquid SXM incubated at 25 °C with constant agitation after inoculation of 4000 spores per ml. Error bars represent standard deviations of the means, and significant differences from wild-type were calculated using Welch’s *t*-tests. The experiment was performed two independent times, each time with two biological replicates (*n* = 4) and three technical replicates. No difference (ns, not significant) was detected in the ability to form conidiospores between *VDH4* and *VDH5* single deletion strains, the *VDH4/5* double deletion strain, and the wild-type strain.

**Figure 4 jof-08-01252-f004:**
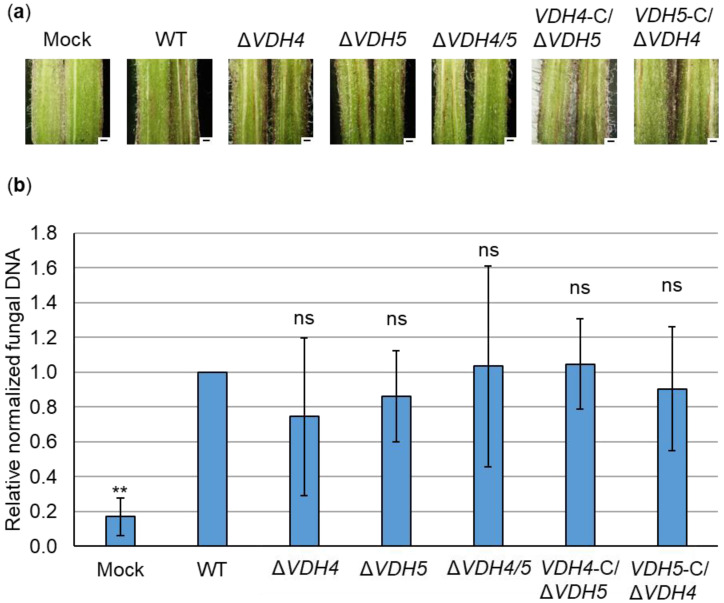
Hydrophobins Vdh4 and Vdh5 are dispensable for colonization of tomato hypocotyls by *V. dahliae*. Tomato seedlings were treated with water as a control (mock) or inoculated with *V. dahliae* spores by root-dipping. JR2 wild-type (WT), *VDH4* (Δ*VDH4*), *VDH5* (Δ*VDH5*), and *VDH4/5* (Δ*VDH4/5*) deletion strains, or *VDH4* (*VDH4*-C/Δ*VDH5*) and *VDH5* (*VDH5*-C/Δ*VDH4*) single gene complementation strains in the double deletion strain background were used. The plants were then incubated in a climate chamber at 16 h: 8 h light: dark at 22–25 °C for 21 days. (**a**) Cross-sections of representative hypocotyls (scale = 500 µm) are shown. Hypocotyl discoloration was visible in plants treated with wild-type, all deletion or complementation strains, but not in mock-inoculated plants. (**b**) Fungal DNA in hypocotyls was analyzed by quantitative PCR relative to plant DNA. Means of four biological replicates from three independent experiments are shown. The error bars represent the standard errors of the mean. Significant differences from wild-type were calculated using Welch’s *t*-tests (ns, not significant; **, *p* < 0.01). Significantly lower levels of fungal DNA were detected only in mock-inoculated hypocotyls but not in plants inoculated with the indicated fungal strains.

**Figure 5 jof-08-01252-f005:**
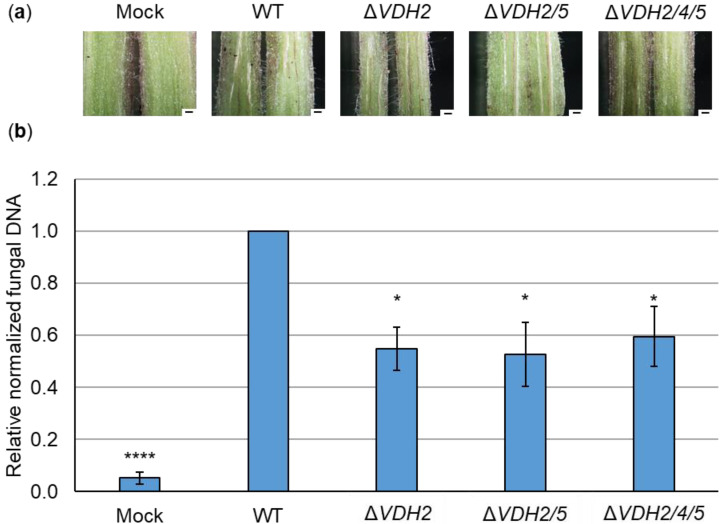
Infection by *V. dahliae VDH2* deletion strains results in less fungal DNA detected in tomato hypocotyls than in wild-type-infected plants. Roots of ten-day-old tomato seedlings were immersed in water as a control (mock) or in spores of JR2 wild-type (WT), *VDH2* single deletion (Δ*VDH2*), *VDH2/5* double deletion (Δ*VDH2/5*), and *VDH2/4/5* triple deletion (Δ*VDH2/4/5*) strains. Plants were incubated in a climate chamber at 16 h: 8 h light: dark at 22–25 °C for 21 days. (**a**) Cross-sections of representative hypocotyls (scale = 500 µm) are shown. Plants treated with wild-type and all deletion strains, but not mock-inoculated plants, exhibited discoloration of the hypocotyl. (**b**) Fungal DNA in hypocotyls was analyzed by quantitative PCR relative to plant DNA. Shown are the mean values of four biological replicates from three independent experiments. The error bars represent the standard errors of the mean. Significant differences from wild-type were calculated using Welch’s *t*-tests (*, *p* < 0.05; ****, *p* < 0.0001). Detected fungal DNA was reduced to about 60% of wild-type levels in plants inoculated with hydrophobin gene deletion strains.

**Figure 6 jof-08-01252-f006:**
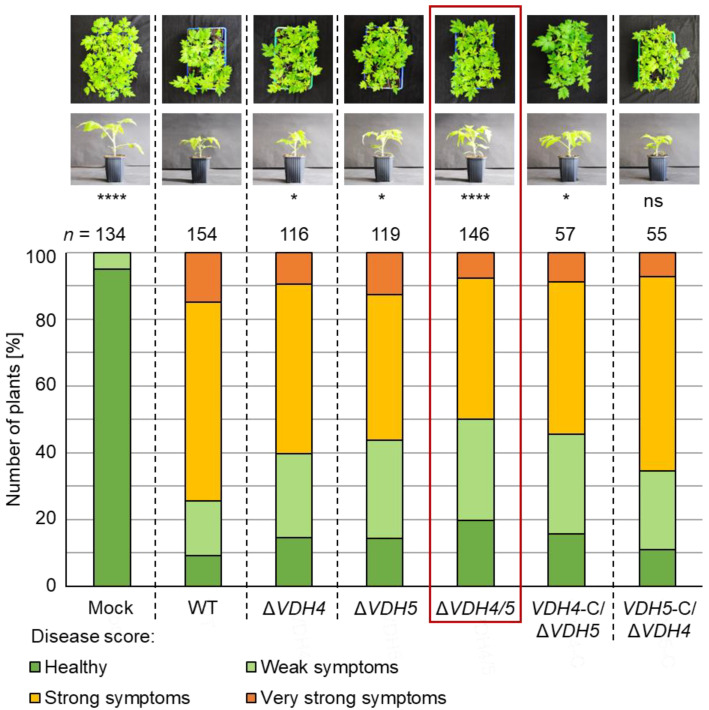
The presence of either *VDH4* or *VDH5* in *V. dahliae* is required for wild-type-like disease development in tomato plants. Tomato seedlings (ten-day-old) were inoculated with *V. dahliae* spores via root-dipping. JR2 wild-type (WT) or *VDH4* (Δ*VDH4*), *VDH5* (Δ*VDH5*), and *VDH4/5* (Δ*VDH4/5*) deletion strains, or *VDH4* (*VDH4*-C/Δ*VDH5*) and *VDH5* (*VDH5*-C/Δ*VDH4*) single gene complementation strains in the double deletion strain background were used. Water-treated plants (mock) served as controls. Plants were incubated in a climate chamber at 16 h: 8 h light: dark at 22–25 °C. A general disease score was determined at 21 days after inoculation based on plant height, length of longest leaf, and plant weight (*n* = number of treated plants). Plant overviews and a single representative plant per treatment are shown. Significant differences from wild-type were calculated using Mann–Whitney U tests. Infection by single deletion strains of *VDH4* or *VDH5* resulted in about 15–20% more plants displaying no or weak rather than strong disease symptoms compared with wild-type-infected plants (*, *p* < 0.05). With simultaneous deletion of both *VDH4* and *VHD5*, this effect was even stronger (****, *p* < 0.0001), as about 25% more plants were healthy or showed only weak disease symptoms (red box). Plants infected by the strain into which *VDH4* was reintroduced (*VDH4*-C/Δ*VDH5*) displayed similar disease symptoms as plants treated with the single deletion strains. Plants infected by the *VDH5*-C/Δ*VDH4* strain were not significantly different from wild-type-infected plants (ns, not significant).

**Figure 7 jof-08-01252-f007:**
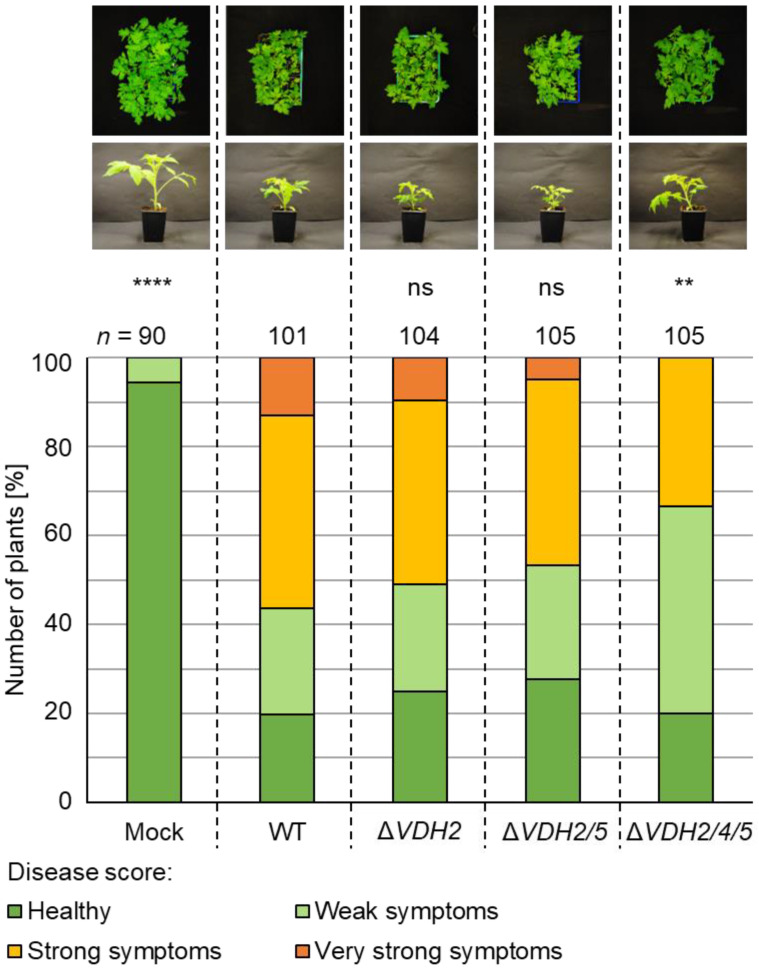
The enhanced triggering of disease symptoms in the presence of both intact *VDH4* and *VDH5* genes is independent of Vdh2. Tomato seedlings (ten-day-old) were infected by spores of JR2 wild-type (WT) or *VDH2* deletion (Δ*VDH2*), *VDH2/5* double deletion (Δ*VDH2/5*), and *VDH2/4/5* triple deletion (Δ*VDH2/4/5*) strains via root-dipping. Water-treated plants (mock) served as controls. Plants were incubated in a climate chamber at 16 h: 8 h light: dark at 22–25 °C. A general disease score was determined at 21 days after inoculation based on plant height, length of longest leaf, and plant weight. The experiment was conducted four independent times (*n* = number of treated plants). Plant overviews and a single representative plant per treatment are shown. Significant differences from wild-type infection were calculated using Mann–Whitney U tests (ns, not significant; **, *p* < 0.01; ****, *p* < 0.0001). Infection by the single or double deletion strains resulted in the same stunting phenotype as infection by the wild-type. Plants treated with the *VDH2/4/5* triple deletion strain did not show very strong symptoms and 25% more plants than wild-type had only weak symptoms.

## Data Availability

The raw RNA sequencing data have been deposited in online repositories (NCBI database, BioProject ID PRJNA881827). Other data presented in this study are available in the Appendix A.

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
