# Peer review of "Tomato Xylem Sap Hydrophobins Vdh4 and Vdh5 Are Important for Late Stages of Verticillium dahliae Plant Infection"

_jof, 2022, doi:10.3390/jof8121252_

Round 1
Reviewer 1 Report
In this study, from RNA sequencing of Verticillium dahliae JR2 in the unfiltered uninfected xylem sap, the authors found four of the five hydrophobins were highly upregulated in the tomato xylem sap. Through comparison of the single mutation, double mutation, triple mutation, complementation, and WT strains of V.dahliae JR2, the study demonstrated that both Vdh4 and Vdh5 are specifically important in the advanced stage of V.dahliae infection in plants and share at least partially overlapping functions. This manuscript is well-organized with a solid experimental design and the conclusion is sufficiently supported by their data. The writing and data presentation are in detail and have clarity.
Here are several minor issues that need to be addressed by the authors:
1. VDH1 single mutation strain is suggested to be included, especially for the evaluation of their effects on the formation of microsclerotia.
2. Please provide the gene accession ID of VDH2, VDH4, and VDH5 in Line 322, like VDH1.
3. Please add a scale bar for figure 3(a)
Author Response
In this study, from RNA sequencing of Verticillium dahliae JR2 in the unfiltered uninfected xylem sap, the authors found four of the five hydrophobins were highly upregulated in the tomato xylem sap. Through comparison of the single mutation, double mutation, triple mutation, complementation, and WT strains of V.dahliae JR2, the study demonstrated that both Vdh4 and Vdh5 are specifically important in the advanced stage of V.dahliae infection in plants and share at least partially overlapping functions. This manuscript is well-organized with a solid experimental design and the conclusion is sufficiently supported by their data. The writing and data presentation are in detail and have clarity.
Here are several minor issues that need to be addressed by the authors:
- VDH1 single mutation strain is suggested to be included, especially for the evaluation of their effects on the formation of microsclerotia.
We included data on a VDH1/4/5 triple mutant (new figure S11a and b, and text page 12, line 450-453 and page 16, line 562-564). Deletion of VDH1 in addition to VDH4/5 did neither change the ability to form microsclerotia in the isolate JR2 under the tested conditions nor did it change the plant infection phenotype we observed for the VDH4/5 double deletion strain.
- Please provide the gene accession ID of VDH2, VDH4, and VDH5 in Line 322, like VDH1.
We inserted the accession IDs as suggested (page 8, line 357-358).
- Please add a scale bar for figure 3(a)
Added as suggested.

Reviewer 2 Report
The manuscript (jof-2011945) analyzed the gene expression of V. dahliae in tomato xylem sap, and provided substantial experimental results, contributing to an in-depth understanding of the function of hydrophobins. Given the transcriptome results is susceptible to the experimental process, the expression of hydrophobins genes could be verified by qRT-PCR.
Author Response
The manuscript (jof-2011945) analyzed the gene expression of V. dahliae in tomato xylem sap, and provided substantial experimental results, contributing to an in-depth understanding of the function of hydrophobins. Given the transcriptome results is susceptible to the experimental process, the expression of hydrophobins genes could be verified by qRT-PCR.
We verified the elevated expression levels of the hydrophobin genes using qRT-PCR as suggested and included them in figure 1 (b) and mentioned it in the text (page 9, line 362-366).
Reviewer 3 Report
This manuscript details the findings of differential gene expression analysis of Verticillium dahliae isolate JR2 during growth in a simulated xylem sap medium versus growth in extracted and slightly diluted xylem sap of tomato. The analysis of this RNAseq data revealed the strong up-regulation of four hydrophobin-encoding genes in addition to eighty or so other genes from various functional classes. Three of these hydrophobin genes were functionally characterized as single, double or triple mutants and in vitro and in planta phenotypes examined. The most significant findings were that two hydrophobins (VDH4 and 5) appear to play important roles in late stages of disease but not in in vitro growth or development. RNAseq and functional gene characterization work are both technically well done and represent solid tools for exploring the biology of V. dahliae. I had hoped to see some cytological work comparing mutant and wild-type plant colonization and a discussion of why mutant phenotypes are only observed in the late stages of disease. In addition to omissions, I also found sections of the discussion that over interpret the presented data. These and other comments are detailed more fully below. Some aspects of the presented work could have been more systematic, e.g., single gene mutants generated for all hydrophobin genes, transcript accumulation profiling during plant infection and full genetic complementation of double mutants, the manuscript represents new data supporting the hypothesis that hydrophobins play a role in Verticillium wilt and open the door for further investigation.
P1L20. The phrase “natural plant environment” appears here to be used to describe extracted xylem sap. This is an oversimplification and potentially misleading to the reader as this isolated sap lack many of the physical attributes of the xylem including the hydraulic dynamics and topography that may also provide signals for gene expression. I suggest changing “natural plant environment” to”xylem sap”
P1 L20-1. The term ‘hydrophobic protein” does not seem to properly describe hydrophobins as these are ampiphilic proteins.
P1L31&34. Again, I question the use of the term “hydrophobic protein” to generically refer to hydrophobins. I do not think it captures the complexity of the physical properties of hydrophobins and also wonder if there are other “hydrophobic proteins” that are not hydrophobins and thus may cause confusion.
P1L41-2. This sentence “The main function of hydrophobins is to confer hydrophobicity to fungal hyphae when they attach to host hydrophobic surfaces, symbiotic partners or themselves” directly copies the list “hydrophobic surfaces, symbiotic partners or themselves” from the cited reference but creates errors of commission and omission. First, the term “host” is added before “hydrophobic surface”. Most fungi do not participate in host interactions and most hydrophobins, even those produced by symbiotic fungi, are not be found in association with host tissues. As such, “the main function of hydrophobins” is put into too narrow a context here. This leads to the second error of omission in which the authors failed to include the fact that most hydrophobins are known to accumulate on the surfaces of fungal tissues (hyphae, spores, etc.) that are in contact with an air interface. One can appreciate the attempt to put the subject matter in context of host-pathogen interactions but the broader biological contexts can not be ignored or down-played and the information in the cited material should be accurately represented.
P2L79. Again the phrase “natural plant environment” is misleading here unless, specific gene transcript accumulation patterns were subsequently analyzed from infected hosts. See also P6L274. Change to “extracted tomato xylem sap” or something similar
P3L98-99. Provide, culture volumes, temperature, rpm
P3L101. “10% sterile distilled water” should be changed to something like “a one tenth volume of sterile distilled water was added”.
P7L314. Change “hydrophobic” to “hydrophobin”
P9L407. Change “hydrophobins” to “hydrophobins VDH2,4 and 5”
P11L442. Why is the VDH2 deletion strain data in the supplemental data instead of included here?
P12L474-482 and Figure 4. Because the whole plant disease assay produces a qualitative and variable result with the wild type and mutants, presentation of data from genetically complemented single deltaVDH4 and delta VDH5 mutant strains and fully complemented double mutant strains, if possible, would be helpful to fully evaluate the mutant phenotypes. The triple mutant phenotype presented in Figure 6 is more convincing but still, lack of genetic complementation data raises the bar for convincing phenotypes. Would it be possible to provide cytological evidence, rate of colonization, biomass or DNA quantification data to strengthen the reported virulence defect phenotypes?
P12L468. Although previous work in another isolate indicated that the loss-of-function mutation did not affect virulence, over-expression of VDH1 did. As this gene exhibited an expression pattern similar to VDH4 and 5 in the current study, it would have been instructive to include it in this study. Why was VDH1 excluded from analysis in this study?
P15L548. This description of the main constituents of xylem sap seems is incredibly over-simplified and perhaps misleading to the reader. It may be appropriate to simply state that the xylem is a physically unique, relatively nutrient poor environment in the plant
P15L554. Change “…by activating the cross-pathway controlled by the transcription factor Cpc1.” To “…by activating the cross-pathway control transcription factor Cpc1.”
P16L581. In the cited study loss-of-function mutation of VDH1 was found to decrease the number of microsclerotia but was not demonstrated to “induce” development. Change“was found to induce the production of micro-sclerotia” to reflect the findings of the cited study.
P16L605. Change “cotton plant disease” to “Verticillium wilt of cotton” or other common name for he disease.
P16L611-612. Change “…detection of wild-type-like fungal DNA amounts…” to something such as “…detection of fungal DNA at levels comparable to the wild type…”
P16L614. Change “hydrophobic” to “hydrophobin”
P16L614. Change “hydrophobin” to “Verticillium hydrophobin”
P16L622. Should “some proteins” be “some hydrophobin proteins”?
P17L638. The statement “the hydrophobins of JR2 are dispensable for development” is a bit too broad as VDH1 and VDH3 were not functionally analyzed in this study. Adjust this statement accordingly.
P17L639. This conclusion “Therefore, overall, our results suggest that hydrophobins play different roles in development and virulence depending on the isolate and host plant.” Is also too broad as one-to-one comparisons of hydrophobin gene mutations have not been made between any two isolates of V. dahliae. Temper this conclusion accordingly.
P17L644. Since the presence of hydrophobin gene transcripts were never checked in planta, the term “natural plant environment” cannot be used here; “extracted xylem sap” may be substituted.
Author Response
This manuscript details the findings of differential gene expression analysis of Verticillium dahliae isolate JR2 during growth in a simulated xylem sap medium versus growth in extracted and slightly diluted xylem sap of tomato. The analysis of this RNAseq data revealed the strong up-regulation of four hydrophobin-encoding genes in addition to eighty or so other genes from various functional classes. Three of these hydrophobin genes were functionally characterized as single, double or triple mutants and in vitro and in planta phenotypes examined. The most significant findings were that two hydrophobins (VDH4 and 5) appear to play important roles in late stages of disease but not in in vitro growth or development. RNAseq and functional gene characterization work are both technically well done and represent solid tools for exploring the biology of V. dahliae. I had hoped to see some cytological work comparing mutant and wild-type plant colonization and a discussion of why mutant phenotypes are only observed in the late stages of disease. In addition to omissions, I also found sections of the discussion that over interpret the presented data. These and other comments are detailed more fully below. Some aspects of the presented work could have been more systematic, e.g., single gene mutants generated for all hydrophobin genes, transcript accumulation profiling during plant infection and full genetic complementation of double mutants, the manuscript represents new data supporting the hypothesis that hydrophobins play a role in Verticillium wilt and open the door for further investigation.
P1L20. The phrase “natural plant environment” appears here to be used to describe extracted xylem sap. This is an oversimplification and potentially misleading to the reader as this isolated sap lack many of the physical attributes of the xylem including the hydraulic dynamics and topography that may also provide signals for gene expression. I suggest changing “natural plant environment” to”xylem sap”
Changed as suggested.
P1 L20-1. The term ‘hydrophobic protein” does not seem to properly describe hydrophobins as these are ampiphilic proteins.
We changed the term according to the reviewers comments.
P1L31&34. Again, I question the use of the term “hydrophobic protein” to generically refer to hydrophobins. I do not think it captures the complexity of the physical properties of hydrophobins and also wonder if there are other “hydrophobic proteins” that are not hydrophobins and thus may cause confusion.
We changed the term according to the reviewers comments.
P1L41-2. This sentence “The main function of hydrophobins is to confer hydrophobicity to fungal hyphae when they attach to host hydrophobic surfaces, symbiotic partners or themselves” directly copies the list “hydrophobic surfaces, symbiotic partners or themselves” from the cited reference but creates errors of commission and omission. First, the term “host” is added before “hydrophobic surface”. Most fungi do not participate in host interactions and most hydrophobins, even those produced by symbiotic fungi, are not be found in association with host tissues. As such, “the main function of hydrophobins” is put into too narrow a context here. This leads to the second error of omission in which the authors failed to include the fact that most hydrophobins are known to accumulate on the surfaces of fungal tissues (hyphae, spores, etc.) that are in contact with an air interface. One can appreciate the attempt to put the subject matter in context of host-pathogen interactions but the broader biological contexts can not be ignored or down-played and the information in the cited material should be accurately represented.
We changed the sentence according to the reviewers comments (page 1-2, line 41-45).
P2L79. Again the phrase “natural plant environment” is misleading here unless, specific gene transcript accumulation patterns were subsequently analyzed from infected hosts. See also P6L274. Change to “extracted tomato xylem sap” or something similar
Changed as suggested.
P3L98-99. Provide, culture volumes, temperature, rpm
Changed as suggested (page 3, line 100-102).
P3L101. “10% sterile distilled water” should be changed to something like “a one tenth volume of sterile distilled water was added”.
Changed as suggested.
P7L314. Change “hydrophobic” to “hydrophobin”
Changed as suggested.
P9L407. Change “hydrophobins” to “hydrophobins VDH2,4 and 5”
Changed as suggested.
P11L442. Why is the VDH2 deletion strain data in the supplemental data instead of included here?
We moved the data to the main text as suggested (figure 5).
P12L474-482 and Figure 4. Because the whole plant disease assay produces a qualitative and variable result with the wild type and mutants, presentation of data from genetically complemented single deltaVDH4 and delta VDH5 mutant strains and fully complemented double mutant strains, if possible, would be helpful to fully evaluate the mutant phenotypes. The triple mutant phenotype presented in Figure 6 is more convincing but still, lack of genetic complementation data raises the bar for convincing phenotypes. Would it be possible to provide cytological evidence, rate of colonization, biomass or DNA quantification data to strengthen the reported virulence defect phenotypes?
We quantified the DNA in the hypocotyl of infected plants (figures 4b). However, we did not observe any significant differences between wildtype infected plants and those infected with the VDH4 and VDH5 single and double deletion strains (page 13, line 495-500). Quantification of fungal DNA in the upper tissue failed as amounts appear to be too small for detection even in wildtype infected plants three weeks after infection.
P12L468. Although previous work in another isolate indicated that the loss-of-function mutation did not affect virulence, over-expression of VDH1 did. As this gene exhibited an expression pattern similar to VDH4 and 5 in the current study, it would have been instructive to include it in this study. Why was VDH1 excluded from analysis in this study?
We included a VDH1/4/5 triple deletion strain in our initial experiments, however, it did neither change the ability to form microsclerotia in the isolate JR2 under the tested conditions nor did it change the plant infection phenotype we observed for the VDH4/5 double deletion strain. We included respective data in additional figure S11a and b and also added respective text (page 12, line 450-453 and page 16, line 562-564).
P15L548. This description of the main constituents of xylem sap seems is incredibly over-simplified and perhaps misleading to the reader. It may be appropriate to simply state that the xylem is a physically unique, relatively nutrient poor environment in the plant
We changed the description according to the reviewers comments (page 17, line 598-599).
P15L554. Change “…by activating the cross-pathway controlled by the transcription factor Cpc1.” To “…by activating the cross-pathway control transcription factor Cpc1.”
Changed as suggested.
P16L581. In the cited study loss-of-function mutation of VDH1 was found to decrease the number of microsclerotia but was not demonstrated to “induce” development. Change“was found to induce the production of micro-sclerotia” to reflect the findings of the cited study.
Changed as suggested.
P16L605. Change “cotton plant disease” to “Verticillium wilt of cotton” or other common name for he disease.
Changed as suggested.
P16L611-612. Change “…detection of wild-type-like fungal DNA amounts…” to something such as “…detection of fungal DNA at levels comparable to the wild type…”
Changed as suggested.
P16L614. Change “hydrophobic” to “hydrophobin”
Changed as suggested.
P16L614. Change “hydrophobin” to “Verticillium hydrophobin”
Changed as suggested.
P16L622. Should “some proteins” be “some hydrophobin proteins”?
Changed as suggested.
P17L638. The statement “the hydrophobins of JR2 are dispensable for development” is a bit too broad as VDH1 and VDH3 were not functionally analyzed in this study. Adjust this statement accordingly.
Changed as suggested.
P17L639. This conclusion “Therefore, overall, our results suggest that hydrophobins play different roles in development and virulence depending on the isolate and host plant.” Is also too broad as one-to-one comparisons of hydrophobin gene mutations have not been made between any two isolates of V. dahliae. Temper this conclusion accordingly.
Changed as suggested.
P17L644. Since the presence of hydrophobin gene transcripts were never checked in planta, the term “natural plant environment” cannot be used here; “extracted xylem sap” may be substituted.
Changed as suggested.